# IOL Power Calculations and Cataract Surgery in Eyes with Previous Small Incision Lenticule Extraction

**DOI:** 10.3390/jcm11154418

**Published:** 2022-07-29

**Authors:** Roman Lischke, Walter Sekundo, Rainer Wiltfang, Martin Bechmann, Thomas C. Kreutzer, Siegfried G. Priglinger, Martin Dirisamer, Nikolaus Luft

**Affiliations:** 1Department of Ophthalmology, University Hospital, Ludwig-Maximilians-University, 80337 Munich, Germany; r.lischke@campus.lmu.de (R.L.); thomas.kreutzer@med.uni-muenchen.de (T.C.K.); siegfried.priglinger@med.uni-muenchen.de (S.G.P.); martin.dirisamer@med.uni-muenchen.de (M.D.); 2SMILE Eyes Clinic, Department of Ophthalmology, Philipps University, 35043 Marburg, Germany; sekundo@med.uni-marburg.de; 3SMILE Eyes Clinic, 85356 Munich, Germany; wiltfang@smileeyes.de (R.W.); bechmann@smileeyes.de (M.B.); 4SMILE Eyes Clinic, 54294 Trier, Germany; 5SMILE Eyes Clinic, 4020 Linz, Austria

**Keywords:** SMILE, IOL calculation, ray tracing, cataract surgery

## Abstract

Small incision lenticule extraction (SMILE), with over 5 million procedures globally performed, will challenge ophthalmologists in the foreseeable future with accurate intraocular lens power calculations in an ageing population. After more than one decade since the introduction of SMILE, only one case report of cataract surgery with IOL implantation after SMILE is present in the peer-reviewed literature. Hence, the scope of the present multicenter study was to compare the IOL power calculation accuracy in post-SMILE eyes between ray tracing and a range of empirically optimized formulae available in the ASCRS post-keratorefractive surgery IOL power online calculator. In our study of 11 post-SMILE eyes undergoing cataract surgery, ray tracing showed the smallest mean absolute error (0.40 D) and yielded the largest percentage of eyes within ±0.50/±1.00 D (82/91%). The next best conventional formula was the Potvin–Hill formula with a mean absolute error of 0.66 D and an ±0.50/±1.00 D accuracy of 45 and 73%, respectively. Analyzing this first cohort of post-SMILE eyes undergoing cataract surgery and IOL implantation, ray tracing showed superior predictability in IOL power calculation over empirically optimized IOL power calculation formulae that were originally intended for use after Excimer-based keratorefractive procedures.

## 1. Introduction

Small incision lenticule extraction (SMILE), with over 5 million procedures performed globally, has evolved to one of the most popular and established keratorefractive procedures for the correction of myopia and myopic astigmatism. In the foreseeable future, the number of patients with prior SMILE treatment requiring cataract surgery is expected to increase accordingly in an ageing population. Inevitably, ophthalmologists will be challenged by accurate intraocular lens (IOL) power calculations for these patients.

There are three major problems in calculating IOL power after any kind of keratorefractive surgery. The first and most significant pitfall lies in the so-called keratometric index error [1]. In traditional keratometry, corneal radii are only measured for the anterior corneal curvature with the posterior corneal curvature radii being empirically extrapolated based on the assumption that the ratio between the anterior and posterior corneal curvature (A/P ratio) is constant, which is not the case after keratorefractive surgery. Secondly, some standard IOL calculation formulae tend to predict a more anterior effective lens position (formula error). Thirdly, the central zone of effective corneal power that had been artificially treated by keratorefractive surgery is estimated from traditional (paracentral) keratometry measurements. Therefore, the corneal power tends to be overestimated (instrument error). All these factors concordantly predispose to an underestimation in required IOL power and therefore lead to dissatisfying hyperopic residual refractive error after IOL implantation in eyes with prior myopic keratorefractive surgery [1].

Several methods have been introduced to address these sources of error and to reduce refractive surprises after keratorefractive surgery [2,3,4,5,6,7,8,9]. New technologies for corneal power measurements that incorporate measurements of the anterior and posterior corneal radii (e.g., total keratometry [10]) were established to enable more accurate predictions. Moreover, sophisticated IOL power calculation formulae have been developed by means of empirical optimization; some of which consider pre-keratorefractive surgery data (Masket [11], Modfied-Masket or Barrett True-K formula), and some do not incorporate any preoperative values (Shammas [12], Barrett True-K no history, Potvin-Hill [13] or Haigis-L formula [14]). Conveniently, a range of these formulae is readily accessible in the American Society of Cataract and Refractive Surgeons (ASCRS) post-keratorefractive surgery IOL power online calculator.

In addition to these empirical formulae, the purely physical ray-tracing approach has demonstrated very good IOL power calculation outcomes in eyes with prior laser in situ keratomileusis (LASIK) or photorefractive keratectomy (PRK) [15,16,17,18]. The ray-tracing method measures the true shape of the cornea after corneal refractive surgery by using the anterior and posterior curvature radii and asphericity of these surfaces. Moreover, the central IOL thickness, the index of refraction and the true geometrical position, as defined by the ACD (distance between the posterior corneal apex and the anterior IOL apex), are used to describe the IOL and calculate its required power accurately. The ray-tracing method also obviates the need for any further historical or clinical data [5,15,16,17,19,20,21]. Ray tracing has been proven to provide reliable and satisfactory results in IOL calculations not only in treatment-naïve eyes but also in eyes after LASIK and PRK [15,16,18,21,22].

After more than one decade since the introduction of SMILE, only one case report of IOL calculation and implantation after SMILE is present in the peer-reviewed literature [23]. However, no formula comparison was reported. Hence, there is a deficiency in postoperative refractive data to optimize existing IOL power calculation formulae for post-SMILE eyes. In addition, corneal aberrometric changes after SMILE are significantly different when compared to the corneal shape changes after femtosecond laser-assisted LASIK (fs-LASIK) [24,25,26], which questions the validity of formulae optimized for Excimer-based photoablative procedures (e.g., the Masket formula) in post-SMILE eyes.

Consequently, the scope of the present multicenter study was to gather the first cohort of post-SMILE patients undergoing cataract extraction with IOL implantation. In this cohort, we set out to compare the refractive prediction error of IOL power calculations between ray tracing and various empirically optimized formulae available in the ASCRS post-keratorefractive surgery IOL power online calculator.

## 2. Materials and Methods

This multicenter cross-sectional study included patients that had previously undergone small incision lenticule extraction (SMILE) for the treatment of myopia and/or myopic astigmatism and later underwent cataract surgery with IOL implantation. The study was conducted at the University Eye Hospital of the Ludwig–Maximilians University (Munich, Germany), the SMILE Eyes Clinic Munich Airport (Munich, Germany) the SMILE Eyes Clinic Trier (Trier, Germany), the SMILE Eyes Center at the Department of Ophthalmology of the University of Marburg (Marburg, Germany) and the SMILE Eyes Clinic Linz (Linz, Austria).

Institutional review board approval of the Ludwig–Maximilians University Munich, was obtained for all aspects of this study; informed consent to use their data for analysis and publication was obtained from all subjects and all study-related procedures adhered to the tenets outlined in the The authors declare no conflict of interest.aration of Helsinki.

### 2.1. SMILE Surgery

All SMILE procedures were performed by highly experienced corneal surgeons utilizing the VisuMax 500-kHz femtosecond laser platform (Carl Zeiss Meditec AG, Jena, Germany) according to the local standards of the participating centers. The intended cap thickness was programmed at 120–130 µm with an intended optical zone size of 6.3 to 6.7 mm in diameter. At the superotemporal position, a single side cut of 50 degrees with a circumferential length of 3.0–4.0 mm was created. No intraoperative or postoperative complications were encountered. The surgical principles of the SMILE technique have been previously described in detail [27].

### 2.2. Cataract Surgery and IOL Implantation

Cataract surgery including IOL selection and implantation was performed by highly experienced corneal surgeons according to the local standards of the participating centers. Standard phacoemulsification with intracapsular IOL implantation was performed in all cases. A 2.5 mm clear incision at the steep corneal axis and two paracenthesis incisions 2 clock hours away towards both directions were created. In case of a toric IOL, a clear corneal incision was created at the temporal corneal aspect. Cohesive and/or dispersive viscoelastic agents were used at the individual surgeon’s discretion. No intraoperative or postoperative complications were encountered. The implanted IOL models and powers are summarized in Table 1.

### 2.3. Subjective Refraction

Subjective manifest refraction was measured using the Jackson cross-cylinder method before and after each procedure. Best-corrected distance visual acuity (CDVA) was determined using standard ETDRS charts at 4 m.

### 2.4. Post-hoc IOL Power Calculation

Post-hoc IOL power calculation was performed utilizing dedicated-ray tracing software (Okulix; Panopsis, Mainz, Germany, Version 9.01) based on preoperative corneal tomography scans (Pentacam HR; Oculus Optikgeräte GmbH, Wetzlar, Germany), and preoperative optical biometry and anterior chamber depth measurements (IOLMaster 500 or 700; Carl Zeiss Meditec AG, Jena, Germany). Moreover, the American Society of Cataract and Refractive Surgery (ASCRS) post-keratorefractive surgery IOL power online calculator (Version 4.9; http://iolcalc.ascrs.org; last accessed on 4 October 2021) was used to calculate the predicted residual refractive error using the following formulae that consider pre-keratorefractive data: Barrett True K, Masket [11] and Modified Masket. Additionally, the following formulae available in the ASCRS calculator were used, which do not incorporate preoperative data: Barrett True K No History, Haigis-L [14], Potvin-Hill [13] and Shammas [12]. In accordance with recent recommendations for IOL power calculation studies [28], no IOL constant optimization was performed but (when appropriate) optimized IOL constants were used as published on the User Group for Laser Interference Biometry (ULIB) website (http://ocusoft.de/ulib/index.htm; last accessed on 4 October 2021).

### 2.5. Statistical Analysis

On the basis of established protocols for studies on IOL power calculation formula accuracy [28,29], the prediction error (PE) was defined as the difference between the actual residual refraction and the residual refraction predicted by the respective IOL power calculation method for the same IOL power and model. The arithmetic mean of the PE was referred to as the mean error (ME). Moreover, all negative errors were converted to positive to calculate the mean absolute error (MAE) as well as the median absolute error (MedAE). Furthermore, the standard deviation, minimum and maximum (range of PE) as well as the percentage of eyes within ±0.50, ±1.00, ±1.50 and ±2.00 diopter (D) are reported [28,29]. Boxplots were created to illustrate the differences in PE between different IOL power calculation formulae. Normal distribution was tested by the Shapiro–Wilk method. The Kruskal–Wallis test was employed to assess the differences in PE between formulae and ray tracing. In addition, the variance of ME was calculated—a smaller variance indicates better consistency of a IOL calculation method [30]. The Fisher’s exact test with the Bonferroni correction was used to test for statistically significant differences between proportions of eyes with PEs within ±0.50 D and ±1.00 D, respectively., A *p*-value of <0.05 was defined as being indicative of statistical significance. All statistical analyses were performed using SPSS 27.0.0.0 for Windows (IBM Corp.; Armonk, NY, USA).

## 3. Results

A total of 11 eyes of 7 patients [1 (14%) female] were included with a mean follow up after SMILE of 2 ± 1 months (range of 1 to 4 months) and a mean follow up after cataract surgery of 8 ± 11 months (range of 1 to 38 months). The mean period of time between SMILE and cataract procedures were 31 ± 16 months (range 12 to 54 months). Subjects’ baseline characteristics are summarized in Table 2. The mean pre- and post-SMILE manifest refraction spherical equivalent (SE) was −5.15 ± 1.31 diopters (D; range: −7.00 to −3.00 D) and −0.48 ± 0.57 D (range: −1.63 to +0.38 D), respectively. The mean preoperative SE before cataract surgery was −2.44 ± 2.48 D (range: −7.63 to +0.63 D). One patient developed a nuclear cataract, which lead to an index myopia of −7.63 D of SE. After cataract surgery, the mean SE amounted to −0.68 ± 0.65 D (range: −2.00 to 0.00 D).

The performance of the investigated IOL power calculation formulae in reference to physical ray tracing is summarized in Table 3 and visualized by boxplots (Figure 1). On average, the formulae concordantly overestimated the required IOL power. Of all investigated traditional formulae, the Potvin–Hill formula yielded the smallest ME (−0.06 ± 0.86 D, range −1.67 to 1.22) and the Shammas formula resulted in the largest IOL power overestimation with a ME of −0.96 ± 1.14 D (range −2.32 to 1.07). Ray tracing was the only method resulting in a hyperopic ME of 0.18 D ± 0.48 D (range −0.43 to 1.22), even though in absolute terms the ME was the lowest (Table 3). Nevertheless, Kruskal–Wallis testing revealed no statistically significant differences in ME between the different IOL power calculation methods (*p* = 0.16).

With respect to MAE and MedAE, ray tracing achieved the smallest MAE (0.40 D) and MedAE (0.36 D) of all examined methods. Of the various tested formulae from the ASCRS calculator, the Potvin–Hill formula yield the smallest MAE (0.66 D), closely followed by the Barrett True-K (0.80 D) and the Masket formula (0.81 D). The Potvin–Hill formula also yielded the smallest MedAE (0.52 D) of all conventional IOL formulae. Kruskal–Wallis testing, however, revealed no statistically significant differences in MAE (*p* = 0.085) and on MedAE (*p* = 0.095). Regarding the variance of ME, the ray-tracing method showed the smallest variance (0.23 D^2^), followed by the Potvin–Hill (0.74 D^2^) and Modified Masket (0.83 D^2^) formulae. The Haigis-L formula showed the highest variance (1.63 D^2^).

With 82%, the ray-tracing method yielded the highest percentage of eyes within a refractive prediction error of ±0.50 D (Figure 2). The next best conventional formula was the Potvin–Hill formula with an ±0.50 D accuracy of 45%. The Haigis-L formula showed the lowest ±0.50 D accuracy of 9%. The Fisher’s exact test indicated significant differences between proportions of eyes with PEs of ±0.50 D (*p* = 0.034). The Bonferroni correction was employed to investigate these differences in detail, showing statistically significant differences between the ray-tracing method and each of the conventional IOL power calculation formulae (all with *p* < 0.001). No statistically significant differences could be found in the proportions of eyes with PEs of ±1.00 D (*p* = 0.754). Nevertheless, the ray-tracing method achieved the highest ±1.00 D accuracy (91%), followed by the Potvin–Hill, Barrett True-K and Barrett True-K no history formulae (all 73%). The Shammas formula showed the lowest ±1.00 D accuracy of 55%.

## 4. Discussion

In this first study of its kind, ray tracing was compared to six established IOL power calculation formulae available in the ASCRS online calculator in post-SMILE eyes undergoing cataract surgery. In our analysis, the ray tracing method showed the most accurate IOL power calculation with a ME of 0.18 ± 0.48 D and 82% of eyes being within ±0.50 D and 91% of eyes within ±1.00 D.

Our findings endorse previous, purely theoretical studies (with no actually performed cataract surgery) in eyes after SMILE. Lazaridis et al. [31] used a theoretical model including virtual IOL implantation to evaluate prediction errors between ray tracing and four conventional IOL power calculation formulae. In their analysis, ray tracing yielded the smallest ME of −0.06 ± 0.40 D and a PE of ±0.5 D in 81.9% of eyes, which is highly coherent with our findings after actual cataract surgery. Moreover, the lowest ME variance (as an indicator of the consistency of an IOL power calculation method), was achieved by ray tracing in both studies. Interestingly, Lazaridis et al. [31] reported better results for the Haigis-L formula (ME of −0.39 ± 0.62 D and 53.4% of eyes with PEs within ± 0.5 D) as compared to our “real world” analysis, where Haigis-L yielded the worst ±0.50 D accuracy of only 9% of all investigated formulae and a ME of −0.81 ± 1.28 D.

In the second previous theoretical study, our group [32] compared the predicted postoperative residual refractive error of the IOL determined by ray tracing with the residual refraction of the same IOL as predicted by a range of conventional IOL power calculation formulae available in the ASCRS post-keratorefractive surgery IOL power calculator. The Masket formula showed the smallest ME (−0.36 ± 0.32 D) and yielded the largest percentage of eyes within ±0.50 D (70%) in reference to the prediction of ray tracing, which was defined as the gold standard method for the purpose of that study. Non-inferior MEs and ±0.50 D accuracies were achieved by the Barrett True K, Barrett True K no history and the Potvin–Hill formula [32].

In the third purely theoretical study, Zhu et al. [33] used the concept of equivalent IOL power differences (EILD) as an indicator for the “stability” of four conventional IOL calculation formulae in post-SMILE eyes. The Barrett True-K and Haigis formulae showed similar stability in eyes with axial lengths between 24 and 26 mm (85.19 vs. 88.89% for a margin of error within 0.5 D; 100 vs. 100% for a margin of error within 1.0 D). In eyes with an axial length of >26 mm, the Barrett True-K formula was the most “stable” formula with respective percentages of 81.49 and 92.59% for margin errors within 0.5 and 1.0 D, respectively.

These compiled theoretical data are confirmed by the present “real world” study, in which the Potvin–Hill and Masket formula showed the best PEs of all conventional formulae. The Potvin–Hill formula yielded the best ME in the present study (ME −0.06 ± 0.86 D and 45% of eyes within ±0.50 D) closely followed by ray tracing (ME 0.18 ± 0.48 D and 82% of eyes within ±0.50 D) and the Masket formula (ME −0.25 ± 0.98 D and 36% of eyes within ±0.50 D). Moreover, the accuracy of the Barrett True K formula was non-inferior when preoperative refractive data were not entered but estimated with the Barrett True K no history formula. By using adjusted keratometry readings, the Shammas formula showed the greatest overestimation of IOL power of all the investigated formulae. Highly congruent findings were also made in the previous theoretical study of our group [32].

As a purely physical approach based on Snell’s law, ray tracing offers many advantages over conventional IOL power calculation formulae in post-keratorefractive surgery eyes. Unlike empirically optimized regression formulae, ray tracing does not rely on any fictional keratometric index or “fudge factors” but utilizes measurements of both the anterior and posterior corneal radii to determine total corneal power. Hence, the need for any empirical optimization, clinical history or preoperative refractive data is obsolete. The latter can be a pivotal advantage in eyes with index myopia due to cataract formation and unknown post-keratorefractive surgery refraction.

These theoretical methodological advantages of the ray tracing principle have been previously proven in different samples of post-Excimer ablation eyes undergoing cataract surgery with IOL implantation [15,16,18]. For instance, Savini et al. [16] yielded 71.4% of 21 post myopic Excimer ablation eyes within ± 0.50 D and 85.7% within ± 1.00 D of the predicted refraction utilizing ray tracing. These results seem comparable to our findings in post-SMILE eyes. Saiki et al. [18] reported slightly subpar outcomes for ray tracing in their sample of 24 post myopic LASIK eyes with ±0.50 D and ±1.00 accuracies of 42 and 75%, respectively. Furthermore, the arithmetic prediction error of ray tracing of 0.63 ± 0.85 D indicated an underestimation of IOL power entailing unpleasant hyperopic residual refractive errors after cataract surgery. In our study, we also observed a minimal hyperopic ME for ray tracing after SMILE, even though it was more than three times smaller (0.18 ± 0.48 D).

First recommendations for clinicians encountering post myopic SMILE patients requiring cataract surgery can be formulated based on the findings of the present study. Physical ray tracing should be employed for IOL power calculation and surgeons should be aware of a slight hyperopic ME of less than +0.25 D when selecting the appropriate IOL power, which is only available in 0.50 D steps for most contemporary IOL models. Ray tracing calculations should ideally be interpreted in conjunction with the Potvin-Hill and Masket formula, which should provide comparable results.

Limitations to this study might be found. First and foremost, the study is limited by its relatively small sample size. Nevertheless, the present work represents the first cohort of post-SMILE patients undergoing cataract surgery and may provide clinicians important guidance for IOL power selection. The paucity of post-SMILE cataract cases in Austria and Germany, where the SMILE technique was developed and first introduced more than a decade ago, also prompted us to include both eyes of some patients into the analysis. For the same reason and due to the multicenter approach, the authors felt inclined to accept multiple IOL types, surgeons and surgical protocols. A further limitation of the present study is that not all formulae currently available in the ASCRS calculator could be included as no Atlas-, Galilei- or OCT-based corneal measurements were available.

In summary, this study comprises the first cohort of post myopic SMILE eyes undergoing cataract surgery and IOL implantation. In post-SMILE eyes, ray tracing facilitated IOL power calculations with a superior accuracy and should be the first choice over conventional IOL power calculation formulae that are empirically optimized for post-Excimer ablation eyes.

## Figures and Tables

**Figure 1 jcm-11-04418-f001:**
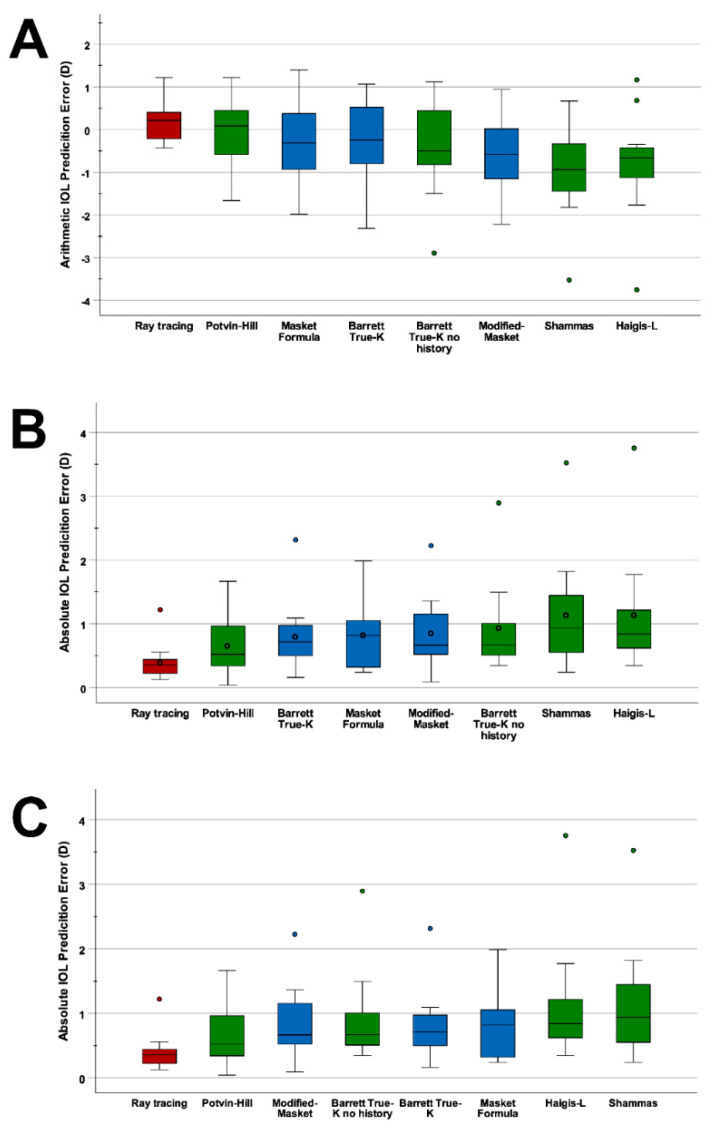
Prediction errors of IOL power calculation formulae. Blue boxplots show formulae that incorporate clinical history data and green boxplots show formulae that do not use any prior keratorefractive surgery data. The red boxplots represent ray tracing. (**A**) IOL power calculation formulae ranked from left to right according to their arithmetic prediction errors. (**B**) IOL power calculation formulae ranked from left to right according to their MAE. Circles demonstrate the respective MAE of each formula. (**C**) IOL power calculation formulae ranked from left to right according to their MedAE. (D, diopter).

**Figure 2 jcm-11-04418-f002:**
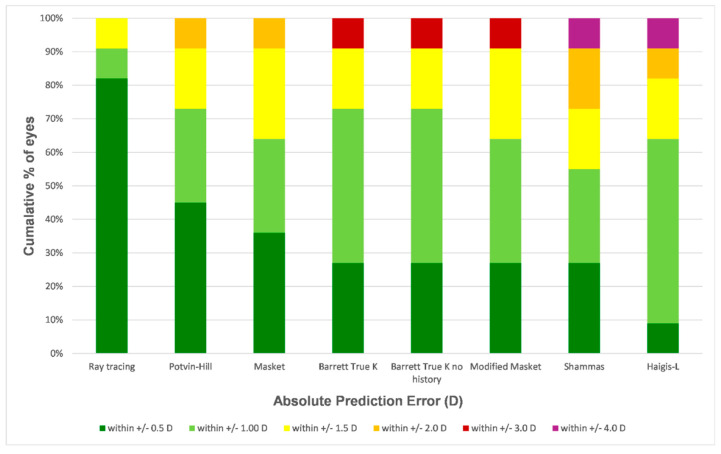
Histogram analysis comparing the percentage of eyes within given prediction error ranges. The formulas were sorted by the proportion of eyes within ±0.50 D in descending order.

**Table 1 jcm-11-04418-t001:** Implanted IOL models, powers and observed prediction errors.

Eye ID	Patient ID	Implanted IOL Model	Manufacturer	IOL Power (Spherical Equivalent, Diopters)	IOL-Power Calculation Formula Used	Prediction Error (Spherical Equivalent, Diopters)
1	1	CT Lucia 601PY	Carl Zeiss Meditec AG (Jena, Germany)	18.5	Haigis-L	0.68
2	1	CT Lucia 601PY	Carl Zeiss Meditec AG (Jena, Germany)	16.5	Haigis-L	−0.67
3	2	CT Lucia 601PY	Carl Zeiss Meditec AG (Jena, Germany)	21.0	Haigis-L	1.17
4	3	CT Lucia 611 PY	Carl Zeiss Meditec AG (Jena, Germany)	22.5	Haigis-L	−1.27
5	4	AcrySof IQ Toric SN6AT2/3	Alcon GmbH (Freiburg, Swiss)	25.0	Haigis-L	−0.51
6	4	AcrySof IQ Toric SN6AT2/3	Alcon GmbH (Freiburg, Swiss)	24.75	Haigis-L	−0.35
7	5	Lentis Comfort LS-313 MF15	Oculentis GmbH (Berlin, Germany)	21.0	Haigis-L	−0.84
8	5	Lentis Comfort LS-313 MF15	Oculentis GmbH (Berlin, Germany)	19.0	Haigis-L	−0.58
9	6	Polylens Y 50 P	Polytech-Domilens GmbH (Roßdorf, Germany)	19.5	Haigis-L	−3.76
10	6	Polylens Y 50 P	Polytech-Domilens GmbH (Roßdorf, Germany)	18.5	Haigis-L	−1.77
11	7	CT Asphina 409 MP	Carl Zeiss Meditec AG (Jena, Germany)	22.0	Ray tracing	−0.62

IOL, intraocular lens; D, diopter.

**Table 2 jcm-11-04418-t002:** Subjects’ characteristics.

Parameter			Mean	Median	SD	Range
Age at SMILE (years)			46.43	46	6.75	37 to 55
Age at cataract surgery (years)			49.45	49	7.31	38 to 59
**SMILE**	Preoperative Manifest Refraction (D)					
		Sphere	−4.86	−5.25	1.30	−6.50 to −2.75
		Cylinder	−0.57	−0.50	0.23	−1.00 to −0.25
		Spherical Equivalent	−5.15	−5.38	1.31	−7.00 to −3.00
	Postoperative Manifest Refraction (D)					
		Sphere	−0.34	−0.5	0.5	−1.75 to 0.50
		Cylinder	−0.27	−0.25	0.24	−0.75 to 0.00
		Spherical Equivalent	−0.48	−0.50	0.57	−1.63 to 0.38
**Cataract surgery**	Preoperative Manifest Refraction (D)					
		Sphere	−2.00	−1.5	2.49	−7.00 to 1.25
		Cylinder	−0.89	−1.00	0.58	−2.00 to −0.25
		Spherical Equivalent	−2.44	−2.25	2.48	−7.63 to 0.63
	Postoperative Manifest Refraction (D)					
		Sphere	−0.45	0.00	0.72	−2.00 to 0.25
		Cylinder	−0.45	−0.5	0.4	−1.25 to 0.00
		Spherical Equivalent	−0.68	−0.63	0.65	−2.00 to 0.00

SD, standard deviation; D, diopter; SMILE, small incision lenticle extraction; BCVA, best corrected visual acuity.

**Table 3 jcm-11-04418-t003:** Formula performance in comparison.

Formula		Prediction Error (D)				Absolute Error (D)		% of Eyes within PE Range Indicated			
		Mean	SD	Range	Variance (D^2^)	Mean	Median	±0.5 D	±1.0 D	±1.5 D	±2.0 D
Ray tracing		0.18	0.48	−0.43 to 1.22	0.23	0.4	0.36	82	91	100	100
Using prior data	Masket	−0.25	0.98	−1.99 to 1.4	0.95	0.81	0.82	36	64	91	100
	Modified-Masket	−0.55	0.91	−2.23 to 0.94	0.83	0.85	0.67	27	64	91	91
	Barret True-K	−0.27	0.98	−2.32 to 1.07	0.96	0.80	0.72	27	73	91	91
Using no prior data	Shammas	−0.96	1.14	−2.53 to 0.67	1.3	1.14	0.94	27	55	73	91
	Haigis-L	−0.81	1.28	−3.76 to 1.17	1.63	1.14	0.84	9	64	82	91
	Potvin-Hill	−0.06	0.86	−1.67 to 1.22	0.74	0.66	0.52	45	73	91	100
	Barrett True K no history	−0.44	1.13	−2.90 to 1.12	1.27	0.93	0.67	27	73	91	91

D, diopters; PE, prediction error; SD, standard deviation.

## Data Availability

The datasets used and analyzed during the current study are available from the corresponding author on reasonable request.

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
