# Peer review of "IOL Power Calculations and Cataract Surgery in Eyes with Previous Small Incision Lenticule Extraction"

_jcm, 2022, doi:10.3390/jcm11154418_

Round 1

Reviewer 1 Report

1)IT IS AN OBSERVATIONAL, STUDY OF 11 EYES

WHAT EXACTLY HAPPENS TO THE CORNEA, KERATOMETRY, KERATOMETRIC INDEX, FRONYT TO BACK RATIO AFTER SMILE?

2) DO YOU HAVE STUDIES TO SHOW THE CHANGES ON THE CURVATURE OF CORNEA AFTER SMILE?

3)LARGER NUMBER IS NEEDE TO CONFIRM

Reviewer 2 Report

The authors present a very nice study on IOL power calculation in patients after SMILE surgery. It is very well written and thorough. There are just a few points I would like some clarification:

Line 155: Why was mean followup time after SMILE only 2 months? Were these patients discharged?

Line 157: Mean time between SMILE and catarct surgery was 31 months ranging from 12 to 54. Did any of the patients have any sign of cataract at the time of the SMILE procedure? I find the development of a visually significant cataract in a little over 2.5 years surprisingly fast. Moreover, table 2 shows the median age for cataract surgery was 49 years. Were these cataract surgeries or refractive lens exchanges?

Line 195: (0.83D2). Change to (0.83D2)

Reviewer 3 Report

Some minor changes:

-I would add a graph synthesizing which formula is better (in a decreasing order) according to the MAE, MedAE and PE so that it is more visual than just the table attached by the authors. *

-In table 1 I would add the formula used by the authors for each case and the EP of each one of them

For the rest of the manuscript, I strongly agree with the authors.

*doi: 10.1177/1120672120980690. PMID: 33339479.

doi: 10.1097/j.jcrs.0000000000000248. PMID: 33060475.

Author Response

This is a much needed and important article for our ophthalmology community. I've

been waiting for a similar article to appear for a long time. For this reason, I would like

to congratulate the authors for such a beautiful job.

1) I would add a graph synthesizing which formula is better (in a decreasing order) according to the MAE, MedAE and PE so that it is more visual than just the table attached by the authors.

We thank the reviewer for these suggestions. We created two additional boxplot graphs to better visualize our data. The IOL power calculation formulae are now ranked from left to right according to their MAE, MedAE and PE, respectively in the novel Figure 1a-c.

2) In table 1 I would add the formula used by the authors for each case and the EP of each one of them.

We thank the reviewer for this valuable comment. We expanded Table 1 as proposed.

3) We would like to point out that the reference cited by the reviewer1 inspired us to amend Figure 2 into a stacked histogram plot for better visualization of data.

References

  1. Rocha-de-Lossada C, Colmenero-Reina E, Flikier D, Castro-Alonso FJ, Rodriguez-Raton A, García-Madrona JL, Peraza-Nieves J, Sánchez-González JM. Intraocular lens power calculation formula accuracy: Comparison of 12 formulas for a trifocal hydrophilic intraocular lens. Eur J Ophthalmol. 2021 Nov;31(6):2981-2988. doi: 10.1177/1120672120980690. Epub 2020 Dec 18. PMID: 33339479.

Reviewer 4 Report

The authors compare the IOL power calculation accuracy in post SMILE eyes between ray tracing and a range of empirically optimized formulae available in the ASCRS post refractive surgery IOL power calculation and they found that ray tracing showed superior predictability in IOL power calculation over empirically optimized IOL power calculation formulae that were originally intended for use after excimer. 

The topic of this manuscript is for interest of refractive surgeon. Post SMILE cataract were incredibly unique since is a ten years technique and not apt to presbyopic, so poor results could be found about this issues. 

Despite the fact that the topic was very novel, the sample size of the study is extremely poor and is impossible to not criticize this issue. In a common IOL power calculation article more than hundreds patients were evaluated since several formulae were included in the analysis comparison. As a reviewer and clinical scientific I know the difficult to find post SMILE cataract, but maybe it is very earlier to make this manuscript and the sample size should be higher in order to confirm and establish the results and to elaborate these conclusions. 

As general comment for future submissions with high eyes number, please shortened the introduction and establish it in a three main paragraph and finish with the purpose clearly state in a separate line. 

In the methods, the SMILE surgery was extremely poor described. I know that all of these manuscripts always repeat the same information on the surgery description, but there were several details to establish in every kind of surgery that could be different from other previous publication. 

Similar comment to cataract and IOL implantation

The heterogeneity of the IOL is other bad points, try to use comparable lens in future research when possible. 

Tables and figures do not follow JCM recommendations

All manuscript with refractive surgery IOL results should included the standard graph from Dann Reinstein published by Journal of Refractive Surgery and Journal of Cataract and Refractive Surgery where I supposed this article were previously rejected. 

Round 2

Reviewer 4 Report

See previous comments